# Experiences and perceptions of meals on wheels volunteers in providing nutritional care to older adults: A qualitative evidence synthesis

Christine Fitzgerald[1,2], Brenda Gabriela Muñoz González[1], Pedro Salinas Escárcega[1], Anne Griffin [1,2]*

1 School of Allied Health, Faculty of Education and Health Sciences, University of Limerick, Limerick, Ireland, 2 Ageing Research Centre, Health Research Institute, University of Limerick, Limerick, Ireland

* anne.griffin@ul.ie

## Abstract

In the community, Meals on Wheels (MoW) programmes are instrumental in the delivery of meals to nutritionally vulnerable older adults. This qualitative evidence synthesis aimed to explore the perceptions and experiences among volunteers of MoW services of their role in supporting nutrition care. Scopus, PubMed, CINAHL, Web of Science, Embase, MEDLINE, and PsycINFO were searched employing qualitative data collection and analysis methods. Results were synthesized using Thomas and Harden's three step approach for thematic synthesis. Three articles were included and two themes were identified: (1) complexity in coordinating MoW to ensure service delivery, and (2) the MoW volunteers' perception of their role in providing nutrition care was eclipsed by the social element of their role. While MoW is pivotal to support older adults' independence, challenges like staffing, funding constraints, and limited community awareness of the service persist. Volunteers' express positivity but face concerns about time commitment and replacement recruitment. Exploring MoW volunteers' broader roles in food insecurity is imperative to understand and address the complex dynamics of providing nutrition care and support to older adults.

## Introduction

Malnutrition prevalence is rising due to population aging and is predicted to reach 29.1% by 2080 [1,2]. Globally, between 5 and 10% of community-dwelling older adults are undernourished [3]. Timely nutrition care provided to older adults contributes to their resilience of living with chronic disease, the ageing process and recovery from ill health. The numerous physiological and psychological effects of malnutrition include deterioration in physical and mental function, and translate to lower quality of life, less favourable health outcomes, and thus more frequent and prolonged hospital admissions [2,4,5]. Malnutrition can emerge slowly and silently in community-dwelling older adults and is challenging to detect [2,6]. A timely identification of malnutrition risk in community-based older adults could prevent poor health outcomes. Food insecurity, defined as 'the inability to consume an adequate quality or sufficient quantity of food in socially acceptable ways, or the uncertainty that one will be

**Data availability statement:** All relevant data are within the manuscript and its Supporting Information files.

**Funding:** The author(s) received no specific funding for this work.

**Competing interests:** The authors have declared that no competing interests exist.

able to do so' [7], has been widely investigated in community-living older adults as a significant contributor to malnutrition [8,9]. In the US, food insecurity among older adults has been reported to have increased significantly, from 5.5% to 12.4% over a 10-year period [10]. Across Europe, food insecurity is seen as significantly more prevalent in women, older adults, those living in one-person and lone-parent households, lower-educated respondents, and those with disabilities [11].

Social and community service agencies play a key role in supporting and sustaining older adults experiencing food insecurity through initiatives that provide necessary resources, support networks, and advocacy efforts tailored to this populations specific needs [12,13]. The key role of social and community service agencies in tackling food insecurity for older adults has been well documented in previous research [8,9,14]. Coordinated, long term support systems allow older adults to remain in their own homes, especially if programmes can target the unmet food and nutrition needs among those at high-risk of malnutrition [12]. Key outcomes for community food and nutrition programmes include decreased risk of malnutrition, prevention and reversal of unintended weight loss, and improved food insecurity [12].

The role of volunteers is central to facilitating social and community service agencies who tackle food insecurity among older adults. In many cases, volunteers dedicate their time, both formally and informally, without seeking payment to be involved in initiatives that address nutrition care and malnutrition [15]. These volunteers can be involved in assistance programs in community settings, such as meal delivery, grocery shopping or chaperoned shopping services, and providing support with cooking, lunch clubs, peer support, transportation, counselling, and lifestyle advice for older adults. Recently, the role of volunteers has been recognized in malnutrition screening and intervention programmes, such as the Patient Association Nutrition Checklist, to help identify and support at-risk individuals [16,17].

In the community, Meals on Wheels (MoW) programmes are instrumental in the delivery of meals to nutritionally vulnerable older adults, enabling them to remain living in their own homes [18,19]. Volunteers in MoW services offer an opportunity for early recognition and intervention to manage malnutrition as they are in constant contact with at-risk older adults [20]. A previous review reported that globally, the terms of purpose and delivery of MoW services are evolving to better meet the needs of older adults residing in the community. It also highlighted the importance of incorporating the views of staff and volunteers who facilitate these changes [21]. The aim of this study is to systematically review and synthesize evidence from qualitative studies that explore the perceptions and experiences of community voluntary service providers regarding their role in supporting nutrition care for older adults through MoW services.

## Materials and methods

Thematic analysis is a prominent research synthesis method, alongside meta-ethnography and meta-synthesis. The synthesis process comprises of three interconnected stages [22,23]. Identified 'free codes' are organized into cohesive clusters, forming 'descriptive' themes. Subsequently, the evolution of 'analytical' themes occurs, emphasizing a progression from initial coding to a more refined and insightful level of analysis. The ENTREQ statement guided the reporting of the stages of this qualitative synthesis (S1 Table; [24]). This synthesis was registered in PROSPERO with ID number CRD42022354425.

### Search strategy

In August 2022, B.M.G. and P.S.E., conducted a scoping search with the help of A. G. to develop the search strategy. Four key terms were retrieved from this initial search:

"Community voluntary providers", "Nutrition care", "Older adults" and "Qualitative". The SPIDER (Sample, Phenomenon of Interest, Design, Evaluation, Research type) framework outlined in Table 1 was used as the foundation for the developing key search concepts [25]. The search was updated in August 2023.

Boolean operator terms were then applied, and the final search was conducted across seven databases: Scopus, PubMed, CINAHL, Web of Science, Embase, MEDLINE, and PsycINFO (Table 2). There was no restriction on publication date for this search. Each database was searched from its origin to the present (S2 Table).

## Screening process

A total of 6702 results were identified from electronic databases and exported into the Rayyan platform (https://rayyan.qcri.org/) to expedite the initial screening [26]. A total of 1979 duplicate records were deleted, and 4724 records were screened. Four researchers (A.G., C.F., B.M.G., P.S.E.) independently screened 25% of the retrieved articles by title and abstract. Forty-six articles were sought for retrieval as at least 1 researcher classified them in the 'maybe' category, C.F. and B.M.G. reviewed 23 articles and A.G. and P.S.E reviewed the remaining 23 articles. Subsequently, 37 articles were identified for full text screening (S3 Table). Any conflicts between the researchers about a study's eligibility were resolved by a third researcher. In cases where full-text publications were unavailable, authors were contacted and asked for full-text versions. The PRISMA diagram (Fig 1) represents the screening and inclusion process [27].

## Eligibility criteria

This review focuses on primary research, using qualitative methods or mixed-methods with separate reporting of qualitative findings, that report the perceptions and experiences of volunteers providing meal support to community-dwelling individuals aged 65 years and older. Inclusion criteria include peer-reviewed articles published in English and Spanish, examining concepts related to nutrition care delivered by MoW services. Volunteer or service provider roles could involve paid, voluntary, or a combination of both. Exclusion criteria included studies conducted in developing countries, descriptions of

**Table 1. SPIDER Framework for search strategy of the research question (Cooke et al. 2012).**

| | |
|---|---|
| **Sample** | Those persons who deliver meals to older adults at home/in community. They might offer this service voluntarily, on retention, or salaried or a combination. The meal delivery service may be part of a healthcare system (e.g., healthcare assistance), community scheme (e.g., meals on wheels), or charitable donations (e.g., food hampers). These services could be implemented from public or private facilities.<br>Given the reviews focus, i.e., the community volunteer who deliver meals, we will exclude the perceptions and experience of clients of these services. |
| **Phenomenon of Interest** (to understand the how and why of individual experiences) | Provision of nutrition care, including the awareness, identification and intervention measures reported. This might include observation by food delivery persons of nutrition-related contributory factors (e.g., transportation, access, availability, skills), signs (e.g., poor appetite, swallow difficulty), symptoms (e.g., weight loss, loose fitting jewellery, pallor, health complaints), finance (e.g., contributions, pension, allowances, etc.) among older adults living in community.<br>The settings of these phenomena are the domiciliary home (individual level) or daycare setting in an older person local to their homes (community level). |
| **Design** (to help make decisions about the robustness of the study and analysis) | Interview, focus groups, document analysis and observations |
| **Evaluation** (outcomes are subjective-attitudes, views, etc.) | Experience and Perceptions of nutrition care and service provider/volunteers' individual role in care pathways that support older adults to remain at home. |
| **Research type** | Qualitative, mixed method (where qualitative data analysis can be extracted) including phenomenology, ethnography, grounded theory, case studies, etc. |

**Table 2. Key terms and Boolean operator terms.**

| Original search terms developed August 2022 | | | |
| --- | --- | --- | --- |
| Community voluntary providers | Nutrition Care | Older adults | Qualitative |
| **MeSH** | | | |
| MM Volunteer Workers<br>MM Volunteer Experiences<br>MM Communities<br>MM Community Networks<br>MM Home Health Aides | MM Food Assistance<br>MM Meals-on-Wheels (Saba CCC)<br>MM Meal Preparation<br>MM Meals | MM Frail Elderly<br>MM Aged | MM Qualitative Studies |
| **Free text search** | | | |
| "Community volunteers" OR community volunt* OR volunteers OR "voluntary provider" OR "community voluntary provider" OR "Non-governmental organization" OR "NGO" | "Malnutrition" OR "malnourished" OR maln* OR "undernutrition" OR "poor nutrition status" OR "nutrition care" OR "nutrition care pathway" OR "identify malnutrition" OR "detect malnutrition" OR "meal provision" OR "meal preparation" OR grocery OR "grocery shopping" OR "grocery provision" OR "food shopping" | "Elderly" OR "older adult" OR senior* OR geriatric* OR "aged individual" OR "aged people" OR "aging" OR "older people" OR "older individual" OR "over 65" OR "sixty five and over" | Qualitative OR "mixed methods" OR "observational study" OR "survey" OR "case studies" OR "evaluation methods" OR interview* OR "focus group*" OR "naturalistic observation" OR "participant observation" OR "social science research" OR transcript* OR ethnography* OR phenomenol* OR "grounded theory*" OR "purposive sample" OR hermeneutic* OR heuristic* OR "lived experience*" OR narrative*<br>OR "life experience*" OR "life stor*" OR "cluster sample" OR "action research" OR "observational method" OR "content analysis" OR "thematic<br>analysis" OR "narrative analysis" OR "constant comparative method" OR "field stud*" OR "fieldnotes" OR "audio recording" OR "video recording" OR "theoretical sample" OR "discourse analysis" |
| Additional suggested search terms based on update August 2023 | | | |
| "community health" OR "community care" OR "domiciliary care" OR "domiciliary support" | "meals on wheels" OR "home-delivered meals" OR "community meals" | "ageing" OR "home-bound older" OR "house bound older" OR "frailty" | |

service providers engaged in medical provision, and studies focusing on meal provision at central venues (excluding home delivery) such as respite or rehabilitation centres. Additionally, studies involving non-in-person meal delivery methods, such as postal systems, were not considered.

## Data collection and quality appraisal

All researchers independently collected data from the selected studies including: study information, design and research type (author, country, sampling, study type, methods of data collection, year of publication and study's aim); sample (paid/ unpaid); description of meal delivery service (reach, organization, mode); phenomenon of interest (awareness of nutrition concerns, provision of nutrition care, pathway instigated by volunteer); evaluation (perception of involvement of meal delivery, nutrition care training, issues related to food insecurity, financial/ budget considerations); supporting evidence (impact of COVID, limitation and/or risk of bias, supporting quotations, major findings).

The Critical Appraisal Skills Programme (CASP) checklist for qualitative studies in healthcare research is recommended by Cochrane and the World Health Organization (WHO) for use in qualitative evidence synthesis [28–30]. Therefore, it was deemed appropriate for this review. The tool was used by each researcher to independently evaluate the included studies (Table 3). Results of the critical appraisals were discussed, and discrepancies (none occurred) settled with consensus.

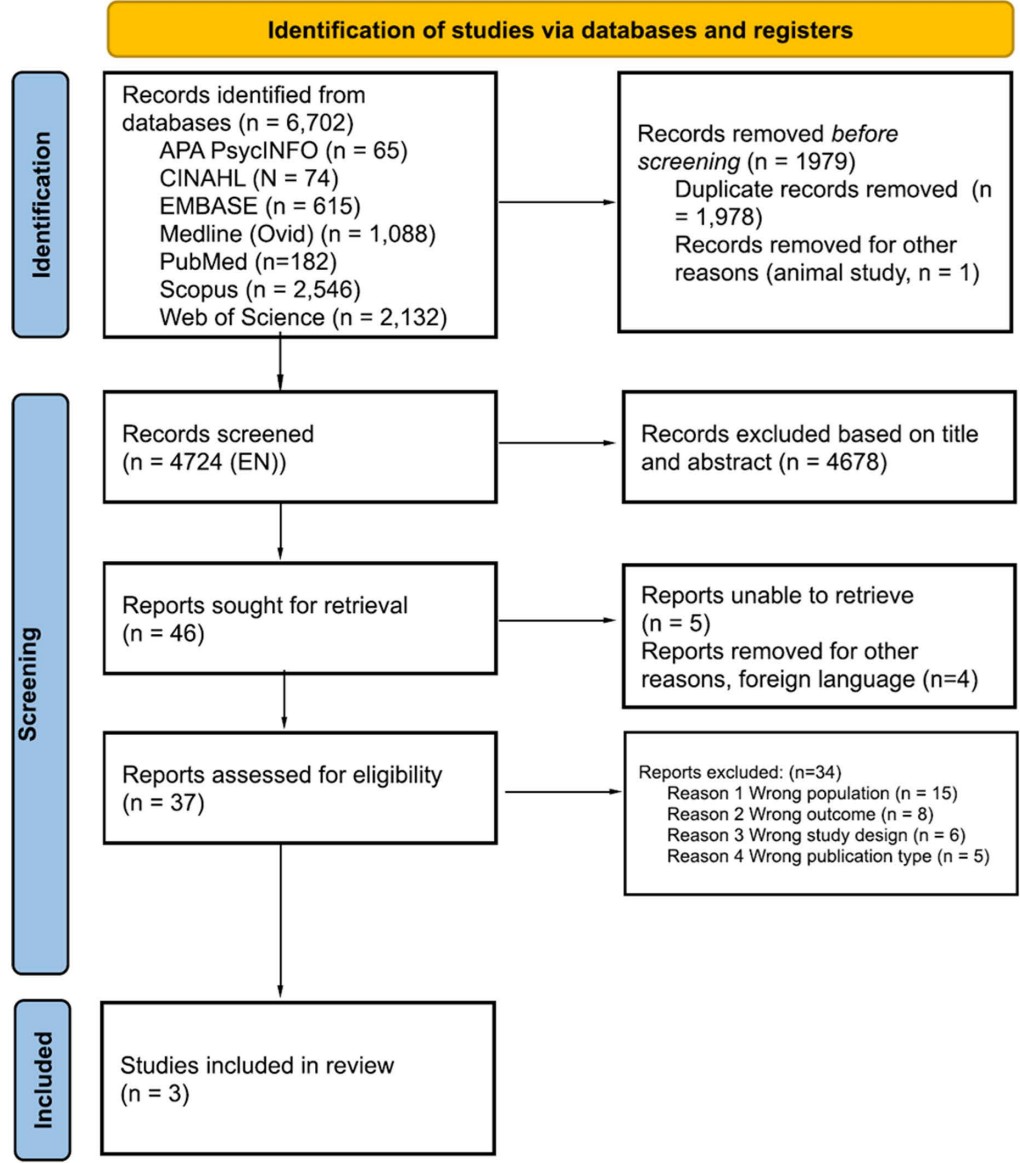

**Fig 1. PRISMA diagram.**

## Data synthesis

Thematic analysis was identified as the most relevant approach for this study, as it is a data analysis technique that enables researchers to condense, emphasize key features, and understand various data sets [31]. It is a useful and effective inductive technique to apply when attempting to comprehend a collection of ideas, feelings, or actions throughout a data set [22,32]. Key concepts to frame the thematic analytical framework of descriptive codes addressing the research question "What are the perceptions and experiences of volunteers supporting the nutrition care of older adults living at home" included:

> *Perception is defined as the way in which nutrition and health needs is regarded, understood, and interpreted by the volunteer.*

**Table 3. CASP qualitative checklist.**

| CASP criteria | O'Dwyer and Timonen 2009 | Papadaki et al 2022 | Thomas et al 2020 |
|---|---|---|---|
| 1. Was there a clear statement of the aims of the research? | Yes | Yes | Yes |
| 2. Is a qualitative methodology appropriate? | Yes | Yes | Yes |
| 3. Was the research design appropriate to address the aims of the research? | Can't tell | Yes | Yes |
| 4. Was the recruitment strategy appropriate to the aims of the research? | Yes | Yes | Yes |
| 5. Was the data collected in a way that addresses the research issue? | Yes | Yes | Yes |
| 6. Has the relationship between researcher and participants been adequately considered? | Can't tell | Can't tell | Yes |
| 7. Have ethical issues been taken into consideration? | Can't tell | Yes | Yes |
| 8. Was the data analysis sufficiently rigorous? | Yes | Can't tell | Yes |
| 9. Is there a clear statement of findings? | Yes | Can't tell | Yes |
| 10. How valuable is the research? | Valuable | Valuable | Valuable |

*Experience is defined as practical contact with and observation of facts/events that leave an impression on the volunteer.*

*Nutrition care is the means of influencing the factors that contribute to imbalance or altered state of nutritional status, e.g., the provision of food and drinks, recognition of poor appetite, etc.*

Initially, the findings from the primary studies underwent free line-by-line coding. Findings can be made from all of the text labelled as 'results' or 'findings' in the study articles [22]. As a first step, A.G. created free line-by-line coding of the verbatim findings (S4 Table) of the selected articles using NVIVO 20, a qualitative data analysis software that allows for analytical practices that supports coding, retrieval of data, and conceptual relationship analysis [33]. Consequently, C.F. revised this extraction and organized the free codes into related areas to construct descriptive themes.

During this stage of analysis, the researchers were mindful of the potential biases stemming from their personal and professional positions. A.G. is a registered dietitian and academic active in the research field of integrated nutrition care and food security of older adults. A.G. is also a steering group member of a collaboration of community, statutory and charity organizations whose aim is to address access, availability and affordability of healthy food in areas of disadvantage. C.F. has a background in nursing and health promotion, and is a postdoctoral researcher focusing on health services research, exploring service evaluation and utilization, client experiences and multi-stakeholder engagement. B.M.G. and P.S.E. were postgraduate students in 2023 who conducted this synthesis towards their MSc Human Nutrition and Dietetics final year project and entry-level registered dietitians working in community in 2024. To ensure rigor, regular meetings were held to discuss descriptive themes in depth and identify analytical themes during this phase of analysis. Analysis took place between 13/11/2023 and 28/03/2024.

## Results

### Descriptive characteristics of the included studies

Each of the three included studies was conducted in a high income country; one in the United States [34] and two in Northwestern Europe ([35]; Ireland); [36]; Britain)). All three studies analyzed data collected by semi-structured interviews. Participants were purposively

sampled and included programme managers [34,35], paid staff [34,36] and volunteers [34,35]. Interviews took place on-site for two studies [34,35] and via telephone for one study [36]. The studies in this review reported on the experiences of 117 participants in total, over half of whom were drivers delivering meals to older adults' homes (n = 68, 58%). The aim of the three included studies aligned with the focus of this synthesis in gathering the perceptions and experiences of meal delivery volunteers. O'Dwyer and Timonen, 2009 aimed to examine the motivations of Irish MoW volunteers and the nature of their contributions in order to explore key considerations and prepare for the future role of volunteering within the service across 13 sites (n = 5 rural and n = 8 urban); Thomas et al, 2021 aimed to describe the interactions of MoW drivers with their clients, and the perceived benefits of the program to the volunteers who make it possible across six sites in the US (n = 4 rural, n = 5 suburban and n = 3 urban); and, Papadaki et al 2021 aimed to explore the perceptions of MoW service providers in England, with regard to the benefits of MoW and the challenges faced by the service in two local authorities (n = 23 urban and n = 20 semi-urban) located in South-West England.

## Theme 1: There is a general experience of complexity in coordinating MoW volunteers to ensure service delivery

The complexity involved in coordinating MoW volunteers was evident in the studies included in this review, highlighting challenges such as volunteer rotas, training for the role, and particularly issues around volunteer recruitment and retention pathways. The coordination of the MoW service included managerial approaches to volunteers, challenges in the day-to-day service delivery, scheduling, training, and recruitment pathways:

*"We fear meals on wheels is in a state of crisis. There is great difficulty in getting funding. There is a great difficulty in recruiting new volunteers" (Coordinator, large rural meals-on-wheels service)*

(O'Dwyer and Timonen, 2009).

MoW management recognized the essential contribution and value of volunteers to the service and acknowledged the sense of burden that some volunteers felt due to their commitments. However, there was variation in how these volunteers were regarded, depending on different managerial approaches. One of the studies, described approaches to coordinating the service as "crisis management" where responses are immediate and contingency plans are put in place. Managers described the importance of showing appreciation to volunteers for the time and effort they provide to ensure the service delivery.

*"Instead, many appeared to use a "crisis-management" style, simply trying to get through each day by dealing with emergencies as they arose, leaving them little time to think about the needs of their volunteers".*

(O'Dwyer & Timonen, 2009).

Coordination efforts faced significant challenges; ongoing pressure in terms of securing funding were underlying across all services. In addition to this challenge, those holding management roles in MoW grappled with an intensive workload, where along with funding concerns, recruitment and retention of volunteers was a major issue creating a sense of uncertainty in the operation of the service.

The demand for additional volunteers and retention of volunteer drivers was reported in the context of these studies as a constant challenge; unusual practices were taken to recruit

new volunteers and organizational protocols were overlooked to make recruitment happen. In some instances, managers reported non- adherence to protocols to retain volunteers and ensure the effective delivery of MoW services to its vulnerable client base, highlighting the pressure to maintain these services.

*"This new approach to recruitment, targeting people not personally known to the coordinator of the service, did not appear to warrant the introduction of new protocols for background checks and other measures for ensuring the safety of both volunteers and clients; none of the coordinators interviewed used formal procedures to "vet" volunteers."*

(O'Dwyer & Timonen, 2009)

Drivers reported being coaxed into volunteering, mirroring the issues of recruitment and retention challenges described from a management perspective. Other challenges included maintaining the motivation of the volunteers in simple meal delivery and the age of volunteers, often older adults themselves.

The managers of MoW describe their workload as multi-faceted including day-to-day tasks of managing the service and future planning in face of funding limitations. It was felt that the funding available does not recognize the role of MoW in older adult care that extends beyond delivery of a meal.

*"Drivers and MOW staff reported that much of the benefit that clients derive from the program is likely attributable to the additional services and support that drivers and the program provide clients. While some drivers did express that their role solely consisted of delivering the meal and exchanging brief pleasantries, the majority of drivers noted providing additional support to their clients beyond delivering the meal."*

(Thomas et al., 2020)

Being a volunteer requires personal sacrifices, like giving up time and other activities. Volunteers described many years of commitment to the MoW service, some of this due to the difficulties of recruiting a replacement driver but also out of recognition of the demands/need for MoW within their community. Existing volunteers played a recruitment role bringing in friends as volunteers. Volunteers are responsible for delivering meals on time, which can limit the social interactions that drivers value. Among the three included studies, there were reported instances of volunteers recognizing a need for additional service and/or emergency responses to older adults in their home. A lack of appropriate training for various aspects of their role, such as promoting physical activity and prompting clients to take prescribed medication, was also noted.

*"We simply don't have the infrastructure to meet the demand... particularly as... Councils have literally been stripped to the bone of funding... now with COVID-19 we can expect probably two, three years down the line even more cuts. I do feel if we can get this on a national level, whether it could be given the respect and appreciation it probably deserves, the meals service (P13/O)."*

(Papadaki et al., 2022)

The challenges facing service coordination did not stop at recruiting volunteers, which showed a wide range of efforts undertaken to recruit new volunteers:

*"How do you go about trying to get voluntary drivers? Oh my goodness, I'd be involved in the church now and I'd be putting things in the newsletter up there, and I'm on another committee and I'm touting for drivers everywhere, everybody I know I'm asking 'have you any time?' Even at bus stops I am asking people!"*

(O'Dwyer & Timonen, 2009)

An ongoing issue was the volunteer pathway. We identified a strong sense of duty of care was among volunteers in the included studies. In some instances, the volunteers reported feeling a sense of responsibility to continue to fulfill the role. Due to the challenges MoW faces in recruiting new volunteers, many current volunteers felt unable to pause their roles and reengage later, resulting in a less than ideal experience for them.

*"An old local man who drove for the organization talked me into doing it, he wanted to give up doing it but he wouldn't give up until he got someone else".*

(O'Dwyer & Timonen, 2009)

The COVID-19 pandemic brought coordination issues to the fore, including use of PPE, the compromised time available to spend with clients, doorstep delivery versus home entry, poor communication, difficult food sourcing, limited human resources and mental and physical health concerns experienced among volunteers.

*"Specific contingency plans included employing casual staff and making sure they had sufficient stocks of PPE."*

(Papadaki et al., 2022)

*"Definitely mood-wise, I get a lot of people now in tears, saying they're lonely, they're frightened. That's hard ."*

(Papadaki et al., 2022)

### Theme 2: MoW volunteers' perception of their role in providing nutrition care was eclipsed by the social element of their role

While the main role of MoW volunteers centered on the delivery of a prepared nutritious meal to an older adult living in their own home, from the articles included in this study, the nutrition role was secondary to the social element of the role.

Volunteers described how they perceived their role as encompassing social support, report, supporting independent living, having an awareness of clients' welfare, and offering additional supports. In addition to providing nutrition care through the delivery of a healthy meal, volunteers recognize their role as providing social support. This includes engaging with older adults and providing social company. A rapport develops to provide a bond between the volunteer and clients, which was something volunteers took a sense of pride in:

*"I also really like the one-on-one interaction; and, I'm very well aware that for some of these folks I'm probably the only human contact they get for a day"*

(Thomas et al., 2020)

*"[but I also] enjoy [volunteering] you'd get to know people and they'd be very chatty"*

(O'Dwyer & Timonen, 2009)

The volunteers described an increased awareness for the welfare of their clients that can provide a health and safety check and respond to emergencies for the older person living at home. In addition, independent living is supported by providing additional practical support to older adults. This can relieve the burden of wider family members.

In addition, benefits of volunteering were described that motivated continued commitment to the MoW service. Volunteers described personal benefits of being involved such as a strong sense of community and satisfaction in "giving back" to a local community.

Beyond the scope of the social and nutrition elements of the role, volunteers also spoke about practical support that they often provided to clients, with examples such as regulating home heating or repairing light fittings:

*"There was sometime during the summer, her [94-year old woman's] apartment was beastly hot, and I kind of waited a few minutes to see if she was going to ask me to help because I didn't want to impose, but then she said, "It's really hot in here, and someone is not helping me with the air conditioning. Will you?" So I did. I was happy about that. I stayed for a few minutes. The air kicked on. I knew it was going to get better for her. I didn't want to leave. I think it was 84 degrees in her apartment"*

(Thomas et al., 2020)

*"Just general: check the gas meter, change the batteries in the clock, put the clock back to the right time. Find someone's glasses. Close windows, open windows… Just small things generally"*

(Papadaki et al., 2022)

In addition to providing practical support, volunteers mentioned a monitoring aspect of their role. As the most consistent source of social contact for clients, MoW volunteers are well positioned to notice any changes or concerns in the health and well-being of clients:

*"Many, many, many times we're the only person some of these people see in a day"*

(Thomas et al., 2020)

*"You get to know all of your people. You see them once a week for months and months and months and months and you know how they're doing, you ask them how they're feeling"*

(Thomas et al., 2020)

This sense of monitoring health and support contributed to a sense of appreciation of the value and self-appreciation offered through their volunteer role:

*"[Clients] feel safe knowing that somebody does come and care, that they really care. And, that gives me a reward in knowing that at the end of the day, that we really save someone.*

*We save them by allowing them to still live independently, and their dignity, you know, is still intact. And they might come to the door with a cane, but they came to the door and it's their door"*

(Thomas et al., 2020)

Despite the challenges faced by MoW volunteers, the awareness of the social value their role provided to clients was sometimes seen as more important than the delivery of a nutritious meal:

*"A lot of people who have not received the service or provided the service thinks it's just a simple meal coming to your home... and it is not just a simple meal. It is so much more. It's that health check, that safety check... making sure that they're okay" (Thomas et al., 2020)*

*"You go in there and you see their faces light up... they are absolutely delighted to see you... because you might be the only person that they see daily"*

(Papadaki et al., 2022)

The social contact that they provided to lone older adults during the pandemic was particularly evident amongst volunteers:

*"Especially during COVID. At least the families were reassured that somebody was going in to check on their relatives and that they were getting something to eat"*

(Papadaki et al., 2022)

There is an opportunity to meet new people with the social contact that delivering meals involves. Some take pride in being able to provide support to the clients. Others describe their commitment as providing them with something to do with their time, particularly at stages of life including retirement from occupational work:

*"On first reading, many of the volunteers, particularly the males, appeared to have volunteered as a result of the lack of alternative activities for occupying their day: And why did you start volunteering on the Meals-on-wheels? When I retired I found that I was home all the time and I got to the stage where I said to the wife one day 'I'm going to go down to the Social Services and see if they want me for anything or other.' It's a case of giving something back rather than what you've got... And I found that they welcomed me with open arms... [the coordinator] said: 'My God, Heaven must have sent you!' (Driver 2) However, such a comment may serve to highlight the sense of satisfaction in feeling needed, particularly at a time of emotional upheaval and readjustment (just following retirement)."*

(O'Dwyer & Timonen, 2009)

## Discussion

### Statement of principal findings

This study aimed to systematically review and synthesize qualitative studies of the perceptions and experiences among MoW volunteers in providing nutritional care to older adults living at home. Our systematic search yielded only three studies, underscoring a notable gap in the evidence. As previously mentioned and evidenced in several systematic reviews, position

statements and empirical studies over the past five years alone, home-delivered meals can effectively provide care and support leading to improved nutrient intake among older adults [9,12,14,37]. However, we identified a focus on the day-to-day logistical elements associated with ensuring older adults received a daily meal against challenges of staffing, funding, and meeting challenges encountered by services. We identified two key themes: complexity in coordinating MoW to ensure service delivery (theme 1), and that MoW volunteers' perception of their role in providing nutrition care was eclipsed by the social element of their role (theme 2).

The MoW service is positively regarded as an essential service to support older adults to remain independent in their homes and communities and maintaining food security for this vulnerable group [34–36]. However, the provision of the MoW service faces some challenges. Our findings provide insights into the complex coordination reported by managers in maintaining service delivery. In the context of the studies reviewed, much of the service delivery described is reactive and unplanned, dependent on day-to-day events and constraints in terms of funding and staffing.

An overall positive sentiment was described among MoW volunteers of being involved in meal delivery, especially where friendships have been forged. Volunteers take great pride in their significant role in maintaining the MoW service, which extended beyond their initial time commitment. They face a conundrum: balancing their personal time spent with the fulfillment of giving back to their community and the joy of using their abilities to help others. These motivations are important factors for positive volunteering experiences [38,39]. Volunteers are more likely to continue their work if they experience positive emotions and feel motivated. Additionally, enjoying their role and feeling part of a team promotes sustained interest in their work [40].

Within the context of the studies reviewed, we observed that the difficulties in maintaining a reliable and quality service due to inadequate and uncertain funding have persisted for over a decade [35,36]. In Ireland, pre-budget submissions have highlighted the precarious nature that MoW service exists within the current climate of rising costs of living: "The current cost of living crisis and rising inflation is putting providers under immense pressure with all across the network reporting increased prices affecting the delivery of their service. Providers are trying to absorb these costs to avoid increasing the cost of their meal to the service user." [41]

Based on the limited data available, MoW volunteers exhibited a lack of awareness regarding their role in providing food security and supporting the well-being of older adults. However, they clearly recognized their role in fostering social interaction among older individuals. This involved coordinating the delivery of nutritious meals to clients, as well as fostering a sense of trust and support from the clients. Furthermore, their involvement provided an avenue for gaining insight and understanding into the intricate care needs of this demographic.

The role of MoW volunteers extends beyond the identification of nutrition malnutrition risk in terms of mitigating against food insecurity among vulnerable older adults. The volunteer role described in the context of this review offers opportunities to understand and address the complex dynamics of providing nutrition care and support to this demographic. Considerations such as navigating a myriad of nutrition challenges such as dietary restrictions, cultural preferences, mobility limitations, and economic constraints require further investigation. Our findings highlight that MoW volunteers prioritize the overall well-being of vulnerable older adults, viewing their primary role as encompassing social interaction, health support, and assistance with practical tasks. Future studies should explore these complex dynamics in greater depth. A mixed-methods approach, combining qualitative interviews with MoW volunteers and quantitative surveys of service users, could provide comprehensive insights into the multifaceted role of volunteers. Additionally, longitudinal studies could track

the long-term impact of MoW services on the nutritional status and overall well-being of older adults, helping to identify best practices and areas for improvement.

Some practical implications to enhance the provision of MoW based on our findings could include enhanced malnutrition training programmes, increased community awareness and policy advocacy. Murphy et al have previously discussed the benefit of having nutrition leads and local nutrition champions to support and empower staff thereby enhancing training programmes that aim to mitigate malnutrition risk in the community [16]. There is a lack of knowledge and awareness about MoW and its services in local communities and among healthcare professionals [25,36] . Family members are most likely to refer service users to MoW during times of need, rather than as a planned or lifestyle choice. This lack of awareness among healthcare professionals, combined with the decision-making process of MoW clients, are important considerations for recruiting new volunteers. To address these challenges, communication efforts and awareness-raising activities could provide a more planned and strategic approach to support volunteer recruitment [38,39]. Finally, advocacy for better policy to support MoW is required. For example, in Ireland there is no legal entitlement to receive or obligation on the state to provide MoW services. Consequently, there is a lack of clear direction and responsibility for supporting and developing this sector, despite the existence of at least 310 individual MoW services across the country [42].

## Strengths and limitations

A strength of this study is the use of robust procedures of thematic analysis to allow a detailed exploration of the available data, identifying underlying themes that add to a more comprehensive understanding of the experience among MoW volunteers engaged in providing nutrition care. This approach enhances the credibility and relevance of the findings, as they are grounded in the lived experiences of volunteers. However, the small number of available studies meeting the synthesis inclusion criteria is a limitation to the generalisation of the findings to all MoW programmes or volunteer populations. Furthermore, the included studies were conducted in high-income countries, and it is uncertain whether the findings would be replicated in lower- and middle-income countries. This indicates a paucity of knowledge on this issue from global areas where the need for MoW service may be significantly higher, that is noted in other systematic reviews synthesizing the evidence of social and economic factors and malnutrition or the risk of malnutrition in older adults [1]. Thematic analysis involves subjective interpretation by researchers, which may introduce bias into the synthesis process. Researchers' backgrounds, experiences, and perspectives can influence how data are analyzed and interpreted, potentially affecting the reliability and validity of the findings.

## Conclusion

While MoW is acknowledged for maintaining independence and supporting older adults to stay in their own homes, its service delivery is hindered by challenges such as staffing, funding constraints, and limited awareness of its role within communities. Despite the positive sentiments expressed by volunteers towards their involvement, concerns regarding time commitment and replacement recruitment persist. Understanding the multifaceted roles of MoW volunteers, particularly in promoting social interaction and overall well-being among older adults, may support implementation of best practice enhanced nutrition care and warrants further investigation.

## Supporting information

**S1 Table. ENTREQ checklist.**
(DOCX)

**S2 Table. Search strategy development.**
(DOCX)

**S3 Table. Reasons for exclusion at full text screening.**
(DOCX)

**S4 Table. Codebook-meals on wheels QES.**
(DOCX)

## Author contributions

**Conceptualization:** Anne Griffin.

**Data curation:** Christine Fitzgerald, Brenda Gabriela Muñoz González, Pedro Salinas Escárcega, Anne Griffin.

**Formal analysis:** Christine Fitzgerald, Brenda Gabriela Muñoz González, Pedro Salinas Escárcega, Anne Griffin.

**Investigation:** Christine Fitzgerald, Brenda Gabriela Muñoz González, Pedro Salinas Escárcega, Anne Griffin.

**Methodology:** Christine Fitzgerald, Brenda Gabriela Muñoz González, Pedro Salinas Escárcega, Anne Griffin.

**Project administration:** Anne Griffin.

**Supervision:** Christine Fitzgerald, Anne Griffin.

**Validation:** Christine Fitzgerald, Brenda Gabriela Muñoz González, Pedro Salinas Escárcega, Anne Griffin.

**Visualization:** Christine Fitzgerald, Brenda Gabriela Muñoz González, Pedro Salinas Escárcega, Anne Griffin.

**Writing – original draft:** Christine Fitzgerald, Brenda Gabriela Muñoz González, Pedro Salinas Escárcega, Anne Griffin.

**Writing – review & editing:** Christine Fitzgerald, Brenda Gabriela Muñoz González, Pedro Salinas Escárcega, Anne Griffin.

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
