## [Decision Letter · Decision Letter 0]

30 Aug 2024

PONE-D-24-27592Experiences and perceptions of Meals on Wheels Volunteers in providing nutritional care to older adults: A Qualitative Evidence SynthesisPLOS ONE

Dear Dr. Griffin,

Thank you for submitting your manuscript to PLOS ONE. After careful consideration, we feel that it has merit but does not fully meet PLOS ONE’s publication criteria as it currently stands. Therefore, we invite you to submit a revised version of the manuscript that addresses the points raised during the review process.

We look forward to receiving your revised manuscript.

Kind regards,

Rabie Adel El Arab

Academic Editor

PLOS ONE

Journal Requirements:

3. We note that this manuscript is a systematic review or meta-analysis; our author guidelines therefore require that you use PRISMA guidance to help improve reporting quality of this type of study. Please upload copies of the completed PRISMA checklist as Supporting Information with a file name “PRISMA checklist.

Additional Editor Comments:

Dear Dr. Griffin and Co-authors,

Thank you for submitting your manuscript titled "Experiences and perceptions of Meals on Wheels Volunteers in providing nutritional care to older adults: A Qualitative Evidence Synthesis" to PLOS ONE. Your study addresses a critical area in community health, providing valuable insights into the role of volunteers in supporting nutrition care among older adults. I have few additional comments that you might address/ clarify

1. Interpretation of Volunteer Burden

Potential Overestimation: The manuscript emphasizes the burden on volunteers, particularly in terms of time commitment and the emotional toll of their roles. While it is important to acknowledge these challenges, the manuscript may overemphasize these issues without providing sufficient data to quantify the extent of the burden. The heavy focus on these challenges might lead to an overestimated perception of volunteer strain, potentially overshadowing the positive aspects of volunteering. Please consider balancing this discussion by including more examples of the positive outcomes and personal fulfillment that volunteers experience.

2. Scope of Impact on Nutrition Care

Potential Underestimation: The manuscript suggests that the nutritional care aspect of MoW volunteers’ roles is secondary to the social elements of their work. While the social aspect is undeniably significant, the manuscript may underestimate the importance of the nutritional support provided by volunteers. The lack of detailed exploration into how volunteers perceive and address nutritional risks could lead to an underestimation of their impact on nutrition care. I recommend revisiting the data to ensure that the nutritional aspects are accurately represented and not undervalued.

3. Generalization of Challenges Across All MoW Programs

Potential Overestimation: The manuscript discusses challenges such as staffing shortages, funding constraints, and volunteer recruitment, and appears to generalize these challenges across all MoW programs based on a small sample of studies. This might lead to an overestimation of the prevalence and severity of these issues, as experiences can vary significantly depending on the location and structure of the MoW program. Please be cautious in making broad statements and consider qualifying your conclusions with phrases such as "based on the limited data available" or "in the context of the studies reviewed."

4. Limited Data Leading to Broad Conclusions

Potential Misinterpretation: Given that the synthesis is based on only three studies, broad conclusions about the experiences of all MoW volunteers might be misinterpreted or overextended. The manuscript could be at risk of overgeneralizing findings from a limited data set, which may not fully represent the diversity of volunteer experiences in different regions or under different operational models. I suggest acknowledging this limitation more explicitly in your discussion and avoiding overly broad generalizations.

5. Thematic Synthesis and Subjectivity

Potential Bias: The thematic synthesis, while systematic, is subject to the researchers’ interpretations. The risk of subjective bias in coding and theme development could lead to either an overestimation or underestimation of certain aspects of the volunteers' experiences, depending on the researchers' perspectives and focus. To mitigate this, please include a discussion of how potential biases were addressed during the synthesis process, such as by ensuring inter-coder reliability or involving multiple researchers in the coding process.

6. Enhancing Methodological Rigor

Clarification of Methodology and Data Synthesis: The manuscript would benefit from a more detailed description of the search strategy, inclusion/exclusion criteria, and the thematic analysis process. This would enhance transparency and allow readers to better understand the scope of your review and the robustness of your findings.

7. Discussion and Conclusion

Practical Implications: The discussion section reiterates the challenges identified but lacks specific, actionable recommendations. I suggest expanding this section to include practical implications for MoW program administrators, policymakers, and volunteers. This could involve suggesting strategies to improve volunteer recruitment and retention, or ways to better integrate nutritional care into the volunteer role.

Recommendations for Future Research: While the need for further research is mentioned, it would be beneficial to outline specific research questions or areas that future studies should explore. For example, investigating the long-term impact of MoW volunteers on nutritional outcomes for older adults, or the effectiveness of different volunteer support models, could be valuable directions for future work.

8. Minor Revisions

Consistency and Clarity: Ensure that all terms and phrases are used consistently throughout the manuscript. For example, the terms "MoW" and "Meals on Wheels" should be consistently abbreviated or spelled out. Additionally, clarify any technical terms or jargon to ensure that the manuscript is accessible to a broad audience.

Grammar and Syntax: Please review the manuscript for any grammatical errors or awkward phrasing.

We look forward to receiving your revised manuscript.

Best regards,

Reviewers' comments:

Reviewer's Responses to Questions

**Comments to the Author**

1. Is the manuscript technically sound, and do the data support the conclusions?

Reviewer #1: No

Reviewer #2: Yes

2. Has the statistical analysis been performed appropriately and rigorously? 

Reviewer #1: N/A

Reviewer #2: Yes

3. Have the authors made all data underlying the findings in their manuscript fully available?

Reviewer #1: Yes

Reviewer #2: Yes

4. Is the manuscript presented in an intelligible fashion and written in standard English?

Reviewer #1: No

Reviewer #2: Yes

5. Review Comments to the Author

Reviewer #1: Why focus only on qualitative? I do not think that the end result of 3 papers is good enough. You should have involved some quantitative aspect to it as well which would have made the findings a lot more interesting,

Reviewer #2: Dear Author, Your idea for research is both novel and insightful, and it has given me fresh insights into the topic. Thank you for making such great contributions to the scientific community. I am looking forward to reading more of your studies in the future.

6. PLOS authors have the option to publish the peer review history of their article (what does this mean? ). If published, this will include your full peer review and any attached files.

**Do you want your identity to be public for this peer review?** For information about this choice, including consent withdrawal, please see our Privacy Policy .

Reviewer #1: No

Reviewer #2: No

---

## [Author Response · Author response to Decision Letter 1]

24 Oct 2024

Dear Editorial team and Reviewers

Thank you for taking the time to review and advise on our manuscript. Please find a file attached providing a point-by-point response. We hope that you will agree that the article is sufficiently enhanced by addressing each of the valid points raised. We have tracked these changes in the original manuscript and also attach a clean revised manuscript.

Kind regards,

Anne Griffin, on behalf of all authors

---

## [Decision Letter · Decision Letter 1]

13 Feb 2025

PONE-D-24-27592R1Experiences and perceptions of Meals on Wheels volunteers in providing nutritional care to older adults: A qualitative evidence synthesisPLOS ONE

Dear Dr. Griffin,

Thank you for submitting your manuscript to PLOS ONE. After careful consideration, we feel that it has merit but does not fully meet PLOS ONE’s publication criteria as it currently stands. Therefore, we invite you to submit a revised version of the manuscript that addresses the points raised during the review process.

The response from the reviewers has been, on the whole, positive. The obvious concern is the small sample - just three primary qualitative studies. However, as you have conducted a thorough search of the literature, the paucity of papers reflects the extant literature and cannot be easily changed, other than by recommending that more such studies are conducted. Reviewer 5 raises questions of differences between high and low income countries, and urban and rural areas. As the three existing studies are all in high income countries, I do not know if it possible for you to discuss differences between high and low income countries, but please do comment on whether there might be  rural/urban differences. Reviewer 3 mentions the low quality of the figure, but I believe that is an artifact of the process of compiling a reviewer pdf (sorry!) - the actual figure file looks fine to me.

We look forward to receiving your revised manuscript.

Kind regards,

Steve Zimmerman, PhD

Senior Editor, PLOS One

Journal Requirements:

Reviewers' comments:

Reviewer's Responses to Questions

**Comments to the Author**

1. If the authors have adequately addressed your comments raised in a previous round of review and you feel that this manuscript is now acceptable for publication, you may indicate that here to bypass the “Comments to the Author” section, enter your conflict of interest statement in the “Confidential to Editor” section, and submit your "Accept" recommendation.

Reviewer #3: All comments have been addressed

Reviewer #4: All comments have been addressed

Reviewer #5: All comments have been addressed

Reviewer #6: All comments have been addressed

2. Is the manuscript technically sound, and do the data support the conclusions?

Reviewer #3: Yes

Reviewer #4: Yes

Reviewer #5: Yes

Reviewer #6: Partly

3. Has the statistical analysis been performed appropriately and rigorously? 

Reviewer #3: Yes

Reviewer #4: Yes

Reviewer #5: Yes

Reviewer #6: N/A

4. Have the authors made all data underlying the findings in their manuscript fully available?

Reviewer #3: Yes

Reviewer #4: Yes

Reviewer #5: Yes

Reviewer #6: Yes

5. Is the manuscript presented in an intelligible fashion and written in standard English?

Reviewer #3: Yes

Reviewer #4: Yes

Reviewer #5: Yes

Reviewer #6: Yes

6. Review Comments to the Author

Reviewer #3: The manuscript entitled "Experiences and Perceptions of Meals on wheels volunteers in providing nutritional care to older adults: A qualitative evidence synthesis " is a very informative manuscript and will greatly contribute to the health sector.The manuscript has no major mistakes but the figures are a little blurred and need high resolution.

Reviewer #4: Dear Authors,

Your manuscript is technically sound, and the data supports the conclusions. The statistical analysis has been performed rigorously and appropriately. All data underlying the findings in your manuscript is fully available. Additionally, the manuscript is presented in a clear manner and written in standard English.

Best regards,

Reviewer #5: The manuscript is a sound piece of research on the efficacy of MoW. However, the sample used raises questiones:

1. This work incorporates studies performed in the US, UK, and Ireland. These appear to be high income countries. Thereofore, a research of this magnitude does not adequately assess the validy of MoW which can more or less be evident among lower income nations. So the sample size used does not comprehensively underpin the methodology.

2. Little evidence is detailed as to whether the study compares MoW in urban vs. rural districts.

If little or no data/findings are available to address these points, this should be atleast mentioned.

Reviewer #6: Thank-you for giving me the opportunity to review this paper. It appears that all the previous reviewers' comments have been addressed as substantial revision to the paper has been made. The literature search process is thorough and the findings explored.

The paper reports a review of three reviews with some analysis of the publications reviewed. The decision as to whether this type of paper fits the scope of the journal lies with the editors.

7. PLOS authors have the option to publish the peer review history of their article (what does this mean? ). If published, this will include your full peer review and any attached files.

**Do you want your identity to be public for this peer review?** For information about this choice, including consent withdrawal, please see our Privacy Policy .

Reviewer #3: **Yes: ** Dr. Shah Faisal Mohammad, Associate Professor/Director MEDIX College of Health Management Sciences and Institute of Nursing Chakdara,Pakistan.

Reviewer #4: **Yes: ** Beisan A. Mohammad

Reviewer #5: No

Reviewer #6: No

---

## [Author Response · Author response to Decision Letter 2]

14 Feb 2025

Thank you to the editorial team and reviewers for comments. We believe that we have addressed these in the attached revised documents.

---

## [Decision Letter · Decision Letter 2]

25 Feb 2025

Experiences and perceptions of Meals on Wheels volunteers in providing nutritional care to older adults: A qualitative evidence synthesis

PONE-D-24-27592R2

Dear Dr. Griffin,

We’re pleased to inform you that your manuscript has been judged scientifically suitable for publication and will be formally accepted for publication once it meets all outstanding technical requirements.

Kind regards,

Nicola Diviani

Academic Editor

PLOS ONE

Additional Editor Comments (optional):

Reviewers' comments:

Reviewer's Responses to Questions

**Comments to the Author**

1. If the authors have adequately addressed your comments raised in a previous round of review and you feel that this manuscript is now acceptable for publication, you may indicate that here to bypass the “Comments to the Author” section, enter your conflict of interest statement in the “Confidential to Editor” section, and submit your "Accept" recommendation.

Reviewer #5: All comments have been addressed

2. Is the manuscript technically sound, and do the data support the conclusions?

Reviewer #5: Yes

3. Has the statistical analysis been performed appropriately and rigorously? 

Reviewer #5: Yes

4. Have the authors made all data underlying the findings in their manuscript fully available?

Reviewer #5: Yes

5. Is the manuscript presented in an intelligible fashion and written in standard English?

Reviewer #5: Yes

6. Review Comments to the Author

Reviewer #5: The points addressed by myself in an earlier revision of the manuscript have been adequately addressed in this revision. Therefore, I have no concerns in accepting this work for publication.

7. PLOS authors have the option to publish the peer review history of their article (what does this mean? ). If published, this will include your full peer review and any attached files.

**Do you want your identity to be public for this peer review?** For information about this choice, including consent withdrawal, please see our Privacy Policy .

Reviewer #5: **Yes: ** Aravindan Benjamin

---

## [Editor Report · Acceptance letter]

PONE-D-24-27592R2

PLOS ONE

Dear Dr. Griffin,

I'm pleased to inform you that your manuscript has been deemed suitable for publication in PLOS ONE. Congratulations! Your manuscript is now being handed over to our production team.

Kind regards,

on behalf of

Dr. Nicola Diviani

Academic Editor

PLOS ONE